# Mobile ALOHA:
# Learning Bimanual Mobile Manipulation using Low-Cost Whole-Body Teleoperation

**Zipeng Fu**[*] **Tony Z. Zhao**[*] **Chelsea Finn**
Stanford University [*]project co-leads
https://mobile-aloha.github.io

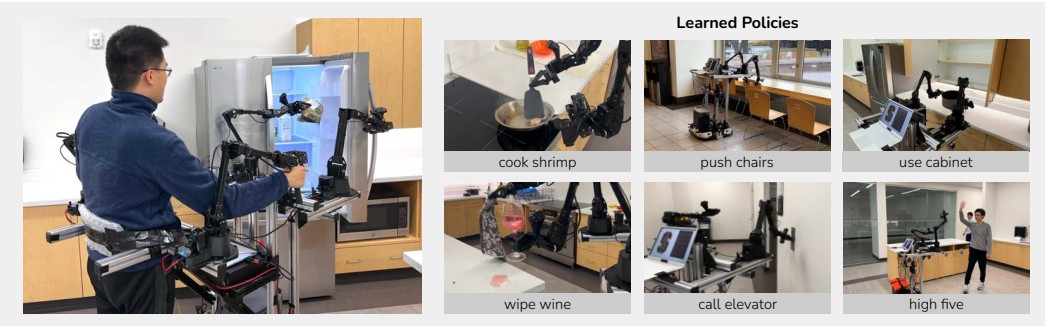

Figure 1: ***Mobile ALOHA***. We introduce a low-cost mobile manipulation system that is bimanual and supports whole-body teleoperation. The system costs $32k including onboard power and compute. *Left:* A user teleoperates to obtain food from the fridge. *Right: Mobile ALOHA* can perform complex long-horizon tasks with imitation learning.

**Abstract:** Imitation learning from human demonstrations has shown impressive performance in robotics. However, most results focus on table-top manipulation, lacking the mobility and dexterity necessary for generally useful tasks. In this work, we develop a system for imitating mobile manipulation tasks that are bimanual and require whole-body control. We first present *Mobile ALOHA*, a low-cost and whole-body teleoperation system for data collection. It augments the *ALOHA* system [1] with a mobile base, and a whole-body teleoperation interface. Using data collected with *Mobile ALOHA*, we then perform supervised behavior cloning and find that co-training with existing *static ALOHA* datasets boosts performance on mobile manipulation tasks. With 50 demonstrations for each task, co-training can increase success rates by up to 90%, allowing *Mobile ALOHA* to autonomously complete complex mobile manipulation tasks such as sauteing and serving a piece of shrimp, opening a two-door wall cabinet to store heavy cooking pots, calling and entering an elevator, and lightly rinsing a used pan using a kitchen faucet.

**Keywords:** Mobile Manipulation, Imitaiton Learning

## 1 Introduction

Imitation learning from human-provided demonstrations is a promising tool for developing generalist robots, as it allows people to teach arbitrary skills to robots. Indeed, direct behavior cloning can enable robots to learn a variety of primitive robot skills ranging from lane-following in mobile robots [2], to simple pick-and-place manipulation skills [3, 4] to more delicate manipulation skills like spreading pizza sauce or slotting in a battery [5, 1]. However, many tasks in realistic, everyday environments require whole-body coordination of both mobility and dexterous manipulation, rather than just individual mobility or manipulation behaviors. For example, consider the relatively basic task of putting away a heavy pot into a cabinet in Figure 1. The robot needs to first navigate to the cabinet, necessitating the mobility of the robot base. To open the cabinet, the robot needs to back up while simultaneously maintaining a firm grasp of the two door handles, motivating whole-body control. Subsequently, both arms need to grasp the pot handles and together move the pot into the cabinet, emphasizing the importance of bimanual coordination. Along a similar vein, cooking,

8th Conference on Robot Learning (CoRL 2024), Munich, Germany.

cleaning, housekeeping, and even simply navigating an office using an elevator all require mobile manipulation and are often made easier with the added flexibility of two arms. In this paper, we study the feasibility of extending imitation learning to tasks that require whole-body control of bimanual mobile robots.

Two main factors hinder the wide adoption of imitation learning for bimanual mobile manipulation. (1) We lack accessible, plug-and-play hardware for whole-body teleoperation. Bimanual mobile manipulators can be costly if purchased off-the-shelf. Robots like the PR2 and the TIAGo can cost more than $200k USD, making them unaffordable for typical research labs. Additional hardware and calibration are also necessary to enable teleoperation on these platforms. For example, the PR1 uses two haptic devices for bimanual teleoperation and foot pedals to control the base [6]. Prior work [7] uses a motion capture system to retarget human motion to a TIAGo robot, which only controls a single arm and needs careful calibration. Gaming controllers and keyboards are also used for teleoperating the Hello Robot Stretch [8] and the Fetch robot [9], but do not support bimanual or whole-body teleoperation. (2) Prior robot learning works have not demonstrated high-performance bimanual mobile manipulation for complex tasks. While many recent works demonstrate that highly expressive policy classes such as diffusion models and transformers can perform well on fine-grained, multi-modal manipulation tasks, it is largely unclear whether the same recipe will hold for mobile manipulation: with additional degrees of freedom added, the interaction between the arms and base actions can be complex, and a small deviation in base pose can lead to large drifts in the arm's end-effector pose. Overall, prior works have not delivered a practical and convincing solution for bimanual mobile manipulation, both from a hardware and a learning standpoint.

We seek to tackle the challenges of applying imitation learning to bimanual mobile manipulation in this paper. On the hardware front, we present *Mobile ALOHA*, a low-cost and whole-body teleoperation system for collecting bimanual mobile manipulation data. *Mobile ALOHA* extends the capabilities of the original *ALOHA* , the low-cost and dexterous bimanual puppeteering setup [1], by mounting it on a wheeled base. The user is then physically tethered to the system and backdrives the wheels to enable base movement. This allows for independent movement of the base while the user has both hands controlling *ALOHA* . We record the base velocity data and the arm puppeteering data at the same time, forming a whole-body teleoperation system.

On the imitation learning front, we observe that simply concatenating the base and arm actions then training via direct imitation learning can yield strong performance. Specifically, we concatenate the 14-DoF joint positions of *ALOHA* with the linear and angular velocity of the mobile base, forming a 16-dimensional action vector. This formulation allows *Mobile ALOHA* to benefit directly from previous deep imitation learning algorithms, requiring almost no change in implementation. To further improve the imitation learning performance, we are inspired by the recent success of pre-training and co-training on diverse robot datasets, while noticing that there are few to none accessible bimanual mobile manipulation datasets. We thus turn to leveraging data from *static* bimanual datasets, which are more abundant and easier to collect, specifically the *static ALOHA* datasets from [1, 10] through the RT-X release [4]. It contains 825 episodes with tasks disjoint from the *Mobile ALOHA* tasks, and has different mounting positions of the two arms. Despite the differences in tasks and morphology, we observe positive transfer in nearly all mobile manipulation tasks, attaining equivalent or better performance and data efficiency than policies trained using only *Mobile ALOHA* data. This observation is also consistent across different class of state-of-the-art imitation learning methods, including ACT [1] and Diffusion Policy [5].

The main contribution of this paper is a system for learning complex mobile bimanual manipulation tasks. Core to this system is both (1) *Mobile ALOHA*, a low-cost whole-body teleoperation system, and (2) the finding that a simple co-training recipe enables data-efficient learning of complex mobile manipulation tasks. Our teleoperation system is capable of multiple hours of consecutive usage, such as cooking a 3-course meal, cleaning a public bathroom, and doing laundry. Our imitation learning result also holds across a wide range of complex tasks such as opening a two-door wall cabinet to store heavy cooking pots, calling an elevator, pushing in chairs, and cleaning up spilled wine. With co-training, we are able to achieve over 80% success on these tasks with only 50 human demonstrations per task, with an average of 34% absolute improvement compared to no co-training.

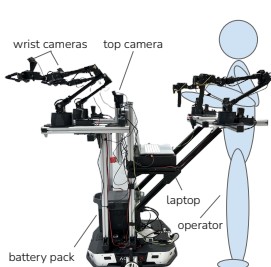 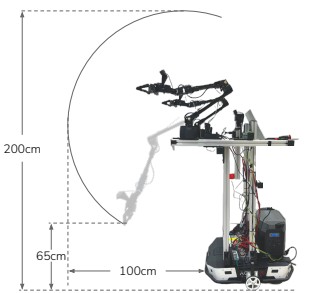

| #Dofs | 14 (arms) + 2 (base) |
|---|---|
| Weight | 75kg |
| Size | 80W * 84L * 140H (no leaders) |
| | 90W * 135L * 140H |
| Payload | 750g (per arm) 55kg (base) |
| Arm Repeatability | 1mm |
| Arm Accuracy | 5-8mm |
| Battery life | 12 hours (1620Wh) |
| Max pulling force | 100N at 100cm vertically |
| Rolling resistance | 13N (vinyl floor) |

Figure 2: **Hardware Details.** *Left: Mobile ALOHA* has two wrist cameras and one top camera, with onboard power and compute. *Middle:* The teleoperation setup can be removed and only two ViperX 300 [11] are used during autonomous execution. Both arms can reach a min/max height of 65cm/200cm, and extends 100cm from the base. *Right:* Technical specifications of *Mobile ALOHA* .

## 2  Related Work

**Mobile Manipulation.** Many current mobile manipulation systems utilize model-based control, which involves integrating human expertise and insights into the system's design and architecture [12, 13, 14, 15, 6]. A notable example of model-based control in mobile manipulation is the DARPA Robotics Challenge [16]. Nonetheless, these systems can be challenging to develop and maintain, often requiring substantial team efforts, and even minor errors in perception modeling can result in significant control failures [17, 18]. Recently, learning-based approaches have been applied to mobile manipulation, alleviating much of the heavy engineering. In order to tackle the exploration problem in high-dimensional state and action spaces of mobile manipulation tasks, prior works use predefined skill primitives [19, 20, 21], reinforcement learning with decomposed action spaces [22, 23, 24, 25, 26], or whole-body control objectives [27, 28, 29]. Unlike these prior works that use action primitives, state estimators, depth images or object bounding boxes, imitation learning allows mobile manipulators to learn end-to-end by directly mapping raw RGB observations to whole-body actions, showing promising results through large-scale training using real-world data [3, 30, 31] in indoor environments [32, 31]. Prior works use expert demonstrations collected by using a VR interface [33], kinesthetic teaching [34], trained RL policies [35], a smartphone interface [36], motion capture systems [7], or from humans [37]. Prior works also develop humanoid teleoperation by using human motion capture suits [38, 39, 40, 41], exoskeleton [42, 43, 44, 45], VR headsets for visual feedbacks [46, 47, 48, 49], and haptic feedback devices [50, 51]. Purushottam et al. develop an exoskeleton suit attached to a force plate for whole-body teleoperation of a wheeled humanoid, However, there is no low-cost solution to collecting whole-body expert demonstrations for bimanual mobile manipulation. We present *Mobile ALOHA* for this problem. It is suitable for hour-long teleoperation, and does not require a FPV goggle for streaming back videos from the robot's egocentric camera or haptic devices.

**Imitation Learning for Robotics.** Imitation learning enables robots to learn from expert demonstrations [2]. Behavioral cloning (BC) is a simple version, mapping observations to actions. Enhancements to BC include incorporating history with various architectures [53, 54, 55, 3], new training objectives [56, 57, 5, 1, 58], regularization [59], motor primitives [60, 61, 62, 63, 64, 65], and data preprocessing [10]. Prior works also focus on multi-task or few-shot imitation learning, [66, 67, 68, 69, 70, 71, 72, 73], language-conditioned imitation learning [74, 75, 55, 3], imitation from play data [76, 77, 78, 79], using human videos [80, 81, 82, 83, 84, 85, 86, 87], and using task-specific structures [88, 89, 75]. Scaling up these algorithms has led to systems adept at generalizing to new objects, instructions, or scenes [90, 55, 3, 91, 92]. Recently, co-training on diverse real-world datasets collected from different but similar types of robots have shown promising results on single-arm manipulation [4, 93, 94, 95, 96], and on navigation [97]. In this work, we use a co-training pipeline for bimanual mobile manipulation by leveraging the existing static bimanual manipulation datasets, and show that our co-training pipeline improves the performance and data efficiency of mobile manipulation policies across all tasks and several imitation learning methods. To our knowledge, we are the first to find that co-training with static manipulation datasets improves the performance and data efficiency of mobile manipulation policies.

# 3 Mobile ALOHA Hardware

We develop *Mobile ALOHA*, a low-cost mobile manipulator that can perform a broad range of household tasks. *Mobile ALOHA* inherits the benefits of the original *ALOHA* system [1], i.e. the low-cost, dexterous, and repairable bimanual teleoperation setup, while extending its capabilities beyond table-top manipulation. Specifically, we incorporate four key design considerations: 1) **Mobile**: The system can move at a speed comparable to human walking, around 1.42m/s. 2) **Stable**: It is stable when manipulating heavy household objects, such as pots and cabinets. 3) **Whole-body teleoperation**: All degrees of freedom can be teleoperated simultaneously, including both arms and the mobile base. 4) **Untethered**: Onboard power and compute.

We choose AgileX Tracer AGV ("Tracer") as the mobile base following *considerations 1 and 2*. Tracer is a low-profile, differential drive mobile base designed for warehouse logistics. It can move up to 1.6m/s similar to average human walking speed. With a maximum payload of 100kg and 17mm height, we can add a balancing weight low to the ground to achieve the desired tip-over stability. We found Tracer to possess sufficient traversability in accessible buildings: it can traverse obstacles as tall as 10mm and slopes as steep as 8 degrees with load, with a minimum ground clearance of 30mm. In practice, we found it capable of more challenging terrains such as traversing the gap between the floor and the elevator. Tracer costs $7,000 in the United States, more than 5x cheaper than AGVs from e.g. Clearpath with similar speed and payload.

We then seek to design a whole-body teleoperation system on top of the Tracer mobile base and *ALOHA* arms, i.e. a teleoperation system that allows simultaneous control of both the base and the two arms (*consideration 3*). This design choice is particularly important in household settings as it expands the available workspace of the robot. Consider the task of opening a two-door cabinet. Even for humans, we naturally step back while opening the doors to avoid collision and awkward joint configurations. Our teleoperation system shall not constrain such coordinated human motion, nor introduce unnecessary artifacts in the collected dataset. However, designing a whole-body teleoperation system can be challenging, as both hands are already occupied by the *ALOHA* leader arms. We found the design of tethering the operator's waist to the mobile base to be the most simple and direct solution, as shown in Figure 2 (left). The human can backdrive the wheels which have very low friction when torqued off. We measure the rolling resistance to be around 13N on vinyl floor, acceptable to most humans. Connecting the operator to the mobile manipulator directly also enables coarse haptic feedback when the robot collides with objects. To improve the ergonomics, the height of the tethering point and the positions of the leader arms can all be independently adjusted up to 30cm. During autonomous execution, the tethering structure can also be detached by loosening 4 screws, together with the two leader arms. This reduces the footprint and weight of the mobile manipulator as shown in Figure 2 (middle). To improve the ergonomics and expand workspace, we also mount the four *ALOHA* arms all facing forward, different from the original *ALOHA* which has arms facing inward.

To make our mobile manipulator untethered (*consideration 4*), we place a 1.26kWh battery that weights 14kg at the base. It also serves as a balancing weight to avoid tipping over. All compute during data collection and inference is conducted on a consumer-grade laptop with Nvidia 3070 Ti GPU (8GB VRAM) and Intel i7-12800H. It accepts streaming from three Logitech C922x RGB webcams, at 480x640 resolution and 50Hz. Two cameras are mounted to the wrist of the follower robots, and the third facing forward. The laptop also accepts proprioception streaming from all 4 arms through USB serial ports, and from the Tracer mobile base through CAN bus. We record the linear and angular velocities of the mobile base to be used as actions of the learned policy. We also record the joint positions of all 4 robot arms to be used as policy observations and actions. We refer readers to the original *ALOHA* paper [1] for more details about the arms.

With design considerations above, we build *Mobile ALOHA* with a $32k budget, comparable to a single industrial cobot such as the Franka Emika Panda. As illustrated in Figure 2 (middle), the mobile manipulator can reach between 65cm and 200cm vertically relative to the ground, can extend 100cm beyond its base, can lift objects that weight 1.5kg, and can exert pulling force of 100N at a height of 1.5m.

We include more technical specifications of *Mobile ALOHA* in Figure 2 (right). Beyond the off-the-shelf robots, we open-source all of the software and hardware parts with a detailed tutorial covering 3D printing, assembly, and software installation.

## 4 Co-training with Static ALOHA Data

The typical approach for using imitation learning to solve real-world robotics tasks relies on using the datasets that are collected on a specific robot hardware platform for a targeted task. This straightforward approach, however, suffers from lengthy data collection processes where human operators collect demonstration data from scratch for every task on the a specific robot hardware platform. The policies trained on these specialized datasets are often not robust to the perceptual perturbations (e.g. distractors and lighting changes) due to the limited visual diversity in these datasets [98]. Recently, co-training on diverse real-world datasets collected from different but similar types of robots have shown promising results on single-arm manipulation [4, 95, 93, 96], and on navigation [97].

In this work, we use a co-training pipeline that leverages the existing *static ALOHA* datasets to improve the performance of imitation learning for mobile manipulation, specifically for the bimanual arm actions. The *static ALOHA* datasets [1, 10] have 825 demonstrations in total for tasks including Ziploc sealing, picking up a fork, candy wrapping, tearing a paper towel, opening a plastic portion cup with a lid, playing with a ping pong, tape dispensing, using a coffee machine, pencil hand-overs, fastening a velcro cable, slotting a battery, and handling over a screw driver. Notice that the *static ALOHA* data is all collected on a black table-top with the two arms fixed to face towards each other. This setup is different from *Mobile ALOHA* where the background changes with the moving base and the two arms are placed in parallel facing the front. We do not use any special data processing techniques on either the RGB observations or the bimanual actions of the *static ALOHA* data for our co-training.

Denote the aggregated *static ALOHA* data as as $D_{\text{static}}$, and the *Mobile ALOHA* dataset for a task $m$ as $D^m_{\text{mobile}}$. The bimanual actions are formulated as target joint positions $a_{\text{arms}} \in \mathbb{R}^{14}$ which includes two continuous gripper actions, and the base actions are formulated as target base linear and angular velocities $a_{\text{base}} \in \mathbb{R}^2$. The training objective for a mobile manipulation policy $\pi^m$ for a task $m$ is

$$\mathbb{E}_{(o^i, a^i_{\text{arms}}, a^i_{\text{base}}) \sim D^m_{\text{mobile}}} \left[ L(a^i_{\text{arms}}, a^i_{\text{base}}, \pi^m(o^i)) \right] \; + \quad \mathbb{E}_{(o^i, a^i_{\text{arms}}) \sim D_{\text{static}}} \left[ L(a^i_{\text{arms}}, [0, 0], \pi^m(o^i)) \right],$$

where $o^i$ is the observation consisting of two wrist camera RGB observations, one egocentric top camera RGB observation mounted between the arms, and joint positions of the arms, and $L$ is the imitation loss function. We sample with equal probability from the *static ALOHA* data $D_{\text{static}}$ and the *Mobile ALOHA* data $D^m_{\text{mobile}}$. We set the batch size to be 16. Since *static ALOHA* datapoints have no mobile base actions, we zero-pad the action labels so actions from both datasets have the same dimension. We also ignore the front camera in the *static ALOHA* data so that both datasets have 3 cameras. We normalize every action based on the statistics of the *Mobile ALOHA* dataset $D^m_{\text{mobile}}$ alone. In our experiments, we combine this co-training recipe with multiple base imitation learning approaches, including ACT [1], Diffusion Policy [5], and VINN [57].

## 5 Tasks

We select 7 tasks that cover a wide range of capabilities, objects, and interactions that may appear in realistic applications. We illustrate them in the Appendix Figure 3. For ***Wipe Wine***, the robot needs to clean up spilled wine on the table. This task requires both mobility and bimanual dexterity. Specifically, the robot needs to first navigate to the faucet and pick up the towel, then navigate back to the table. With one arm lifting the wine glass, the other arm needs to wipe the table as well as the bottom of the glass with the towel. This task is not possible with *static ALOHA*, and would take more time for a single-armed mobile robot to accomplish.

For ***Cook Shrimp***, the robot sautes one piece of raw shrimp on both sides before serving it in a bowl. Mobility and bimanual dexterity are also necessary for this task: the robot needs to move from the stove to the kitchen island as well as flipping the shrimp with spatula while the other arm tilting the pan. This task requires more precision than wiping wine due to the complex dynamics of flipping a half-cooked shrimp. Since the shrimp may slightly stick to the pan, it is difficult for the robot to reach under the shrimp with the spatula and precisely flip it over.

For ***Rinse Pan***, the robot picks up a dirty pan and rinse it under the faucet before placing it on the

| | Wipe Wine (50 demos) | | | | Cook Shrimp (20 demos) | | | | |
|---|---|---|---|---|---|---|---|---|---|
| | Grasp Towel | Lift Glass and Wipe | Place Glass | *Whole Task* | Add Oil | Add Shrimp | Flip Shrimp | Plate Shrimp | *Whole Task* |
| Co-train | 100 | 95 | 100 | **95** | 100 | 100 | 60 | 67 | **40** |
| No Co-train | 95 | 58 | 90 | 50 | 100 | 100 | 40 | 50 | 20 |

| | Rinse Pan (50 demos) | | | | Use Cabinet (50 demos) | | | | |
|---|---|---|---|---|---|---|---|---|---|
| | Grasp Pan | Turn On Faucet | Place Pan | *Whole Task* | Open Cabinets | Grasp Pot | Place Pot | Close Cabinet | *Whole Task* |
| Co-train | 100 | 80 | 100 | **80** | 95 | 100 | 95 | 95 | **85** |
| No Co-train | 100 | 0 | 100 | 0 | 95 | 95 | 100 | 95 | **85** |

| | Call Elevator (50 demos) | | | | Push Chairs (50 demos) | | | | High Five (20 demos) | | | |
|---|---|---|---|---|---|---|---|---|---|---|---|---|
| | Navi. | Press Button | Enter Elevator | *Whole Task* | 1-3rd Chair | 4th (OOD) | 5th (OOD) | *Whole Task* | Unseen Attire | Unseen Human | Navi. | *Whole Task* |
| Co-train | 100 | 100 | 95 | **95** | 100 | 85 | 89 | **80** | 90 | 80 | 100 | **85** |
| No Co-train | 100 | 5 | 0 | 0 | 100 | 70 | 0 | 0 | 90 | 80 | 100 | **85** |

Table 1: *Co-training improves ACT performance*. Across 7 challenging mobile manipulation tasks, co-training with *static ALOHA* dataset consistently improve the success rate (%) of ACT. It is particularly important for sub-tasks like *Press Button* in *Call Elevator* and *Turn on Faucet* in *Rinse Pan*, where precise manipulation is the bottleneck.

drying rack. In addition to the challenges in the previous two tasks, turning on the faucet poses a hard perception challenge. The knob is made from shiny stainless steel and is small in size: roughly 4cm in length and 0.7cm in diameter. Due to the stochasticity introduced by the base motion, the arm needs to actively compensate for the errors by "visually-servoing" to the shiny knob. A centimeter-level error could result in task failure.

For *Use Cabinet*, the robot picks up a heavy pot and places it inside a two-door cabinet. While seemingly a task that require no base movement, the robot actually needs to move back and forth four times to accomplish this task. For example when opening the cabinet door, both arms need to grasp the handle while the base is moving backward. This is necessary to avoid collision with the door and have both arms within their workspace. Maneuvers like this also stress the importance of whole-body teleoperation and control: if the arms and base control are separate, the robot will not be able to open both doors quickly and fluidly. Notably, the heaviest pot in our experiments weighs 1.4kg, exceeding the single arm's payload limit of 750g while within the combined payload of two arms.

For *Call Elevator*, the robot needs to enter the elevator by pressing the button. We emphasize long navigation, large randomization, and precise whole-body control in this task. The robot starts around 15m from the elevator and is randomized across the 10m wide lobby. To press the elevator button, the robot needs to go around a column and stop precisely next to the button. Pressing the button, measured 2cm×2cm in size, requires precision as pressing the peripheral or pressing too lightly will not activate the elevator. The robot also needs to turn sharply and precisely to enter the elevator door: there is only 30cm in clearance between the robot's widest part and the door.

For *Push Chairs*, the robot needs to push in 5 chairs in front of a long desk. This task emphasizes the strength of the mobile manipulator: it needs to overcome the friction between the 5kg chair and the ground with coordinated arms and base movement. To make this task more challenging, we only collect data for the first 3 chairs, and stress test the robot to extrapolate to the 4th and 5th chair.

For *High Five*, the robot needs to go around the kitchen island, and whenever a human approach it from the front, stop moving and high five with the human. After the high five, the robot should continue moving only when the human moves out of its path. We collect data wearing different clothes and evaluate the trained policy on unseen persons and unseen attires. While this task does not require a lot of precision, it highlights *Mobile ALOHA*'s potential for studying human-robot interactions.

We want to highlight that for all tasks mentioned above, open-loop replaying a demonstration with objects restored to the same configurations will achieve zero whole-task success. Successfully completing the task requires the learned policy to react close-loop and correct for those errors. We believe the source of errors during the open-loop replaying is the mobile base's velocity control. As

| | | Wipe Wine (50 demos) | | | | Push Chairs (50 demos) | | | |
|---|---|---|---|---|---|---|---|---|---|
| | | Grasp Towel | Lift Glass and Wipe | Place Glass | *Whole Task* | 1st Chair | 2nd Chair | 3rd Chair | *Whole Task* |
| VINN + Chunking | Co-train | 85 | 18 | 100 | 15 | 100 | 70 | 86 | **60** |
| | No Co-train | 50 | 40 | 100 | **20** | 90 | 72 | 62 | 40 |
| Diffusion Policy | Co-train | 90 | 72 | 100 | **65** | 100 | 100 | 100 | **100** |
| | No Co-train | 75 | 47 | 100 | 35 | 100 | 80 | 100 | 80 |
| ACT | Co-train | 100 | 95 | 100 | **95** | 100 | 100 | 100 | **100** |
| | No Co-train | 95 | 58 | 90 | 50 | 100 | 100 | 100 | **100** |

Table 2: *Mobile ALOHA is compatible with recent imitation learning methods.* VINN with chunking, Diffusion Policy, and ACT all achieves good performance on *Mobile ALOHA*, and benefit from co-training with *static ALOHA*.

an example, we observe $> 10$cm error on average when replaying the base actions for a 180 degree turn with 1m radius.

# 6 Experiments

We aim to answer two central questions in our experiments. (1) Can *Mobile ALOHA* acquire complex mobile manipulation skills with co-training and a small amount of mobile manipulation data? (2) Can *Mobile ALOHA* work with different types of imitation learning methods, including ACT [1], Diffusion Policy [5], and retrieval-based VINN [57] We conduct extensive experiments in the real-world to examine these questions. **Please check the Appendix B and C for ablations and user studies, and our project website for qualitative results and videos.**.

As a preliminary, all methods we will examine employ "action chunking" [1], where a policy predicts a sequence of future actions instead of one action at each timestep. It is already part of the method for ACT and Diffusion policy, and simple to be added for VINN. We found action chunking to be crucial for manipulation, improving the coherence of generated trajectory and reducing the latency from per-step policy inference. Action chunking also provides a unique advantage for *Mobile ALOHA*: handling the delay of different parts of the hardware more flexibly. We observe a delay between target and actual velocities of our mobile base, while the delay for position-controlled arms is much smaller. To account for a delay of $d$ steps of the mobile base, our robot executes the first $k - d$ arm actions and last $k - d$ base actions of an action chunk of length $k$.

## 6.1 Co-training Improves Performance

We start with ACT [1], the method introduced with ALOHA, and train it on all 7 tasks with and without co-training. We then evaluate each policy in the real-world, with randomization of robot and objects configurations as described in Figure 3. To calculate the success rate for a sub-task, we divide $\#Success$ by $\#Attempts$. For example in the case of *Lift Glass and Wipe* sub-task, the $\#Attempts$ equals the number of success from the previous sub-task *Grasp Towel*, as the robot could fail and stop at any sub-task. This also means the final success rate equals the product of all sub-task success rates. We report all success rates in Table 1. Each success rate is computed from 20 trials of evaluation, except *Cook Shrimp* which has 5.

With the help of co-training, the robot obtains 95% success for *Wipe Wine*, 95% success for *Call Elevator*, 85% success for *Use Cabinet*, 85% success for *High Five*, 80% success for *Rinse Pan*, and 80% success for *Push Chairs*. Each of these tasks only requires 50 in-domain demonstrations, or 20 in the case of *High Five*. The only task that falls below 80% success is *Cook Shrimp* (40%), which is a 75-second long-horizon task for which we only collected 20 demonstrations. We found the policy to struggle with flipping the shrimp with the spatula and pouring the shrimp inside the white bowl, which has low contrast with the white table. We hypothesize that the lower success is likely due to the limited demonstration data. Co-training improves the whole-task success rate in 5 out of the 7 tasks, with a boost of 45%, 20%, 80%, 95% and 80% respectively. For the remaining two tasks, the success rate is comparable between co-training and no co-training. We find co-training to be more helpful for sub-tasks where precise manipulation is the bottleneck, for example *Press Button*, *Flip Shrimp*, and *Turn On Faucet*. In all of these cases, compounding errors appear to be the main source of failure, either from the stochasticity of robot base velocity control or from rich contacts such as grasping of the spatula and making contact with the pan during *Flip Shrimp*. We

hypothesize that the "motion prior" of grasping and approaching objects in the *static ALOHA* dataset still benefits *Mobile ALOHA*, especially given the invariances introduced by the wrist camera [99]. We also find the co-trained policy to generalize better in the case of *Push Chairs* and *Wipe Wine*. For *Push Chairs*, both co-training and no co-training achieve perfect success for the first 3 chairs, which are seen in the demonstrations. However, co-training performs much better when extrapolating to the 4th and 5th chair, by 15% and 89% respectively. For *Wipe Wine*, we observe that the co-trained policy performs better at the boundary of the wine glass randomization region. We thus hypothesize that co-training can also help prevent overfitting, given the low-data regime of 20-50 demonstrations and the expressive transformer-based policy used.

## 6.2 Compatibility with ACT, Diffusion Policy, and VINN

We train two recent imitation learning methods, Diffusion Policy [5] and VINN [57], with *Mobile ALOHA* in addition to ACT. Diffusion policy trains a neural network to gradually refine the action prediction. We use the DDIM scheduler [100] to improve inference speed, and apply data augmentation to image observations to prevent overfitting. The co-training data pipeline is the same as ACT. VINN trains a visual representation model, BYOL [101] and uses it to retrieve actions from the demonstration dataset with nearest neighbors. We augment VINN retrieval with proprioception features and tune the relative weight to balance visual and proprioception feature importance. We also retrieve an action chunk instead of a single action and find significant performance improvement similar to [1]. For co-training, we simply co-train the BYOL encoder with the combined mobile and static data.

In Table 2, we report co-training and no cotraining success rates on 2 real-world tasks: *Wipe Wine* and *Push Chairs*. Overall, Diffusion Policy performs similarly to ACT on *Push Chairs*, both obtaining 100% with co-training. For *Wipe Wine*, we observe worse performance with diffusion at 65% success. The Diffusion Policy is less precise when approaching the kitchen island and grasping the wine glass. We hypothesize that 50 demonstrations is not enough for diffusion given its expressiveness: previous works that utilize Diffusion Policy tend to train on upwards of 250 demonstrations. For VINN + Chunking, the policy performs worse than ACT or Diffusion across the board, while still reaching reasonable success rates with 60% on *Push Chairs* and 15% on *Wipe Wine*. The main failure modes are imprecise grasping on *Lift Glass and Wipe* as well as jerky motion when switching between chunks. We find that increasing the weight on proprioception when retrieving can improve the smoothness while at a cost of paying less attention to visual inputs. We find co-training to improve Diffusion Policy's performance, by 30% and 20% for on *Wipe Wine* and *Push Chairs* respectively. This is expected as co-training helps address overfitting. Unlike ACT and Diffusion Policy, we observe mixed results for VINN, where co-training hurts *Wipe Wine* by 5% while improves *Push Chairs* by 20%. Only the representations of VINN are co-trained, while the action prediction mechanism of VINN does not have a way to leverage the out-of-domain *static ALOHA* data, perhaps explaining these mixed results.

## 7 Conclusion, Limitations and Future Directions

In summary, our paper tackles both the hardware and the software aspects of bimanual mobile manipulation. Augmenting the *ALOHA* system with a mobile base and whole-body teleoperation allows us to collect high-quality demonstrations on complex mobile manipulation tasks. Then through imitation learning co-trained with static *ALOHA* data, *Mobile ALOHA* can learn to perform these tasks with only 20 to 50 demonstrations. We are also able to keep the system accessible, with under $32k budget including onboard power and compute, and open-sourcing on both software and hardware.

Despite *Mobile ALOHA*'s simplicity and performance, there are still limitations that we hope to address in future works. On the hardware front, we will seek to reduce the occupied area of *Mobile ALOHA*. The current footprint of 90cm x 135cm could be too narrow for certain paths. In addition, the fixed height of the two follower arms makes lower cabinets, ovens, and dish washers challenging to reach. We are planning to add more degrees of freedom to the arms' elevation to address this issue. On the software front, we limit our policy learning results to single task imitation learning. The robot can not yet improve itself autonomously or explore to acquire new knowledge. In addition, the *Mobile ALOHA* demonstrations are collected by two expert operators. We leave it to future work for tackling imitation learning from highly suboptimal, heterogeneous datasets.

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

# A Task Illustration

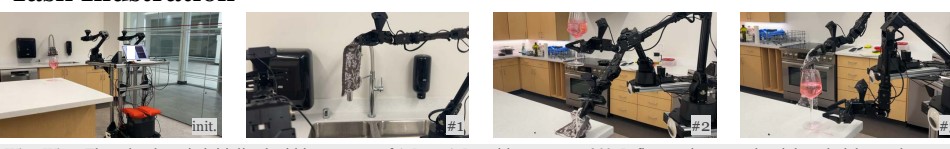

*Wipe Wine*: The robot base is initialized within a square of 1.5m x 1.5m with yaw up to 30°. It first navigates to the sink and picks up the towel hanging on the faucet (#1). It then turns around and approaches the kitchen island, picks up the wine glass (randomized in 30cm x 30cm), wipes the spilled wine (#2), and puts down the wine glass on the table (#3). Each demo has 1300 steps or 26 seconds.

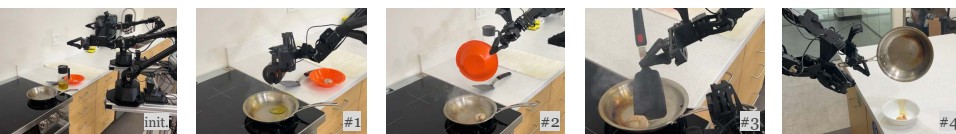

*Cook Shrimp:* The robot is randomized up to 5cm and all objects up to 2cm. The right gripper first pours oil into the hot pan (#1) followed by raw shrimp (#2). With left gripper lifting the pan at an angle, the right gripper grasps the spatula and flips the shrimp (#3). The robot then turns around and pours the shrimp into an empty bowl (#4) before placing the pan on the table. Each demo has 3750 steps or 75 seconds.

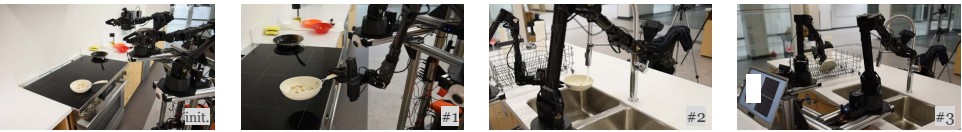

*Rinse Pan:* The pan randomized up to 10cm with yaw up to 45°. The left gripper grasps the pan (#1) before turning around to the faucet. The right gripper opens then closes the faucet with left gripper holding the pan to receive the water (#2). The left gripper then swirls the water inside the pan, pours it out, before placing the pan on the rack (#3). Each demo has 1100 steps or 22 seconds.

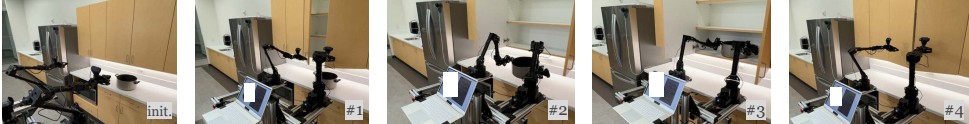

*Use Cabinet:* The robot is randomized up to 10cm and pots up to 5cm. A total of 3 pots are used. The robot first approaches the cabinet and grasp both handles, then backs up pulling both doors open (#1). Next, both arms grasp the handles of the pot, move forward, and place it inside the cabinet (#2). The robot then backs up and closes both cabinet doors (#4). Each demo has 1500 steps or 30 seconds.

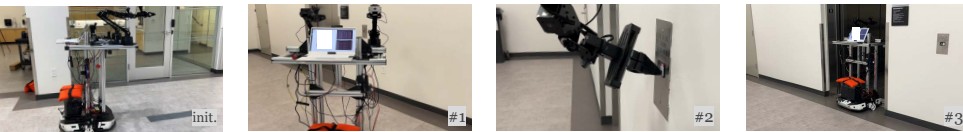

*Call Elevator:* The robot starts 15m from the elevator and is randomized across the 10m wide lobby. The robot goes around a column to reach the elevator button (#1). The right gripper presses the button (#2) and the robot enters the elevator (#3). Each demo has 2250 steps or 45 seconds.

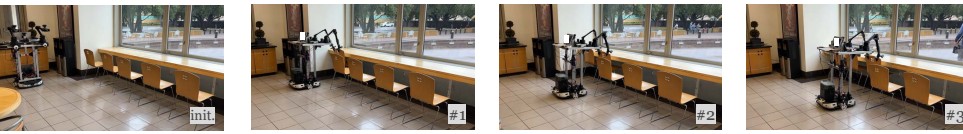

*Push Chairs:* The robot's initial position is randomized up to 10cm. Demonstration dataset contains pushing in the first 3 chairs, and the robot is tested with all 5 chairs. Each demo has 2000 steps or 40 seconds.

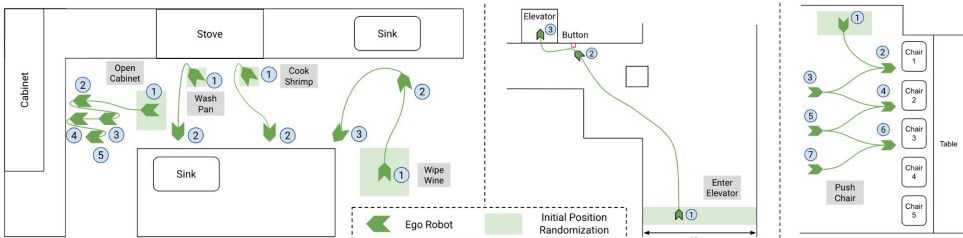

Figure 3: ***Task Definitions.*** We illustrate 6 real-world tasks that *Mobile ALOHA* can perform autonomously. For each task, we describe randomization and sub-task definitions. We also include an illustration of the base movement for each task (not drawn to scale).

# B Ablation Studies

**Data Efficiency.** In Figure 4, we ablate the number of mobile manipulation demonstrations for both co-training and no co-training, using ACT on the *Wipe Wine* task. We consider 25, 35, and 50 *Mobile ALOHA* demonstrations and evaluate for 20 trials each. We observe that co-training leads to better

data efficiency and consistent improvements over training using only *Mobile ALOHA* data. With co-training, the policy trained with 35 in-domain demonstrations can outperform the no co-training policy trained with 50 in-domain demonstrations, by 20% (70% vs. 50%).

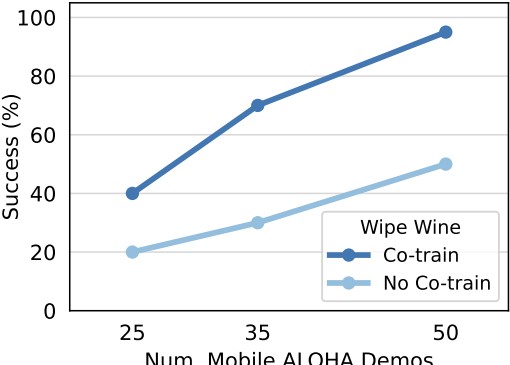

Figure 4: ***Data efficiency.*** Co-training with *static ALOHA* data leads to better data efficiency and consistent improvements over training with *Mobile ALOHA* data only. Figure style credits to [102].

**Co-training Is Robust To Different Data Mixtures.** We sample with equal probability from the *static ALOHA* datasets and the *Mobile ALOHA* task dataset to form a training mini-batch in our co-training experiments so far, giving a co-training data sampling rate of roughly 50%.
In Table 3, we study how different sampling strategies affect performance on the *Wipe Wine* task. We train ACT with 30% and 70% co-training data sampling rates in addition to 50%, then evaluate 20 trials each. We see similar performance across the board, with 95%, 95% and 90% success respectively. This experiment suggests that co-training performance is not sensitive to different data mixtures, reducing the manual tuning necessary when incorporating co-training on a new task.

| Static ALOHA proportion (%) | 30 | 50 (default) | 70 |
|---|---|---|---|
| Success (%) | 95 | 95 | 90 |

Table 3: ***Co-training is robust to different data mixtures.*** Result uses ACT training on the *Wipe Wine* task.

**Co-training Outperforms Pre-training.** In Table 4, we compare co-training and pre-training on the *static ALOHA* data. For pre-training, we first train ACT on the *static ALOHA* data for 10K steps and then continue training with in-domain task data. We experiment with the *Wipe Wine* task and observe that pre-training provides no improvements over training solely on *Wipe Wine* data. We hypothesize that the network forgets its experience on the *static ALOHA* data during the fine-tuning phase.

| | Co-train | Pre-train | No Co-train No Pre-train |
|---|---|---|---|
| Success (%) | **95** | 40 | 50 |

Table 4: ***Co-train vs. Pre-train.*** Co-train outperforms pre-train on the *Wipe Wine* task. For pre-train, we first train ACT on the *static ALOHA* data and then fine-tune it with the *Mobile ALOHA* data.

## C   User Studies

We conduct a user study to evaluate the effectiveness of *Mobile ALOHA* teleoperation. Specifically, we measure how fast participants are able to learn to teleoperate an unseen task. We recruit 8 participants among computer science graduate students, with 5 females and 3 males aged 21-26. Four participants has no prior teleoperation experience, and the remaining 4 have varying levels of

expertise. None of them have used *Mobile ALOHA* before. We start by allowing each participant to freely interact with objects in the scene for 3 minutes. We held out all objects that will be used for the unseen tasks during this process. Next, we give each participants two tasks: *Wipe Wine* and *Use Cabinet*. An expert operator will first demonstrate the task, followed by 5 consecutive trials from the participants. We record the completion time for each trial, and plot them in Figure 5. We notice a steep decline in completion time: on average, the time it took to perform the task went from 46s to 28s for Wipe Wine (down 39%), and from 75s to 36s for Use Cabinet (down 52%). The average participant can also approach speed of expert demonstrations after 5 trials, demonstrating the ease of use and learning of *Mobile ALOHA* teleoperation.

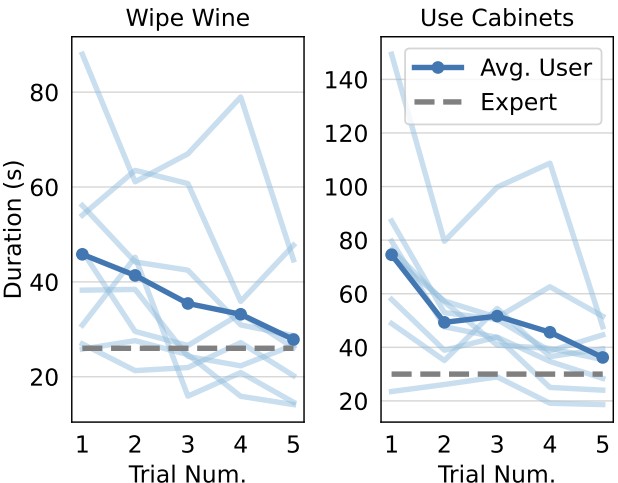

Figure 5: *Teleoperator learning curves.* New users can quickly approach expert speed on an unseen task teleoperating *Mobile ALOHA*.

