# OpenReview forum: "Mobile ALOHA: Learning Bimanual Mobile Manipulation using Low-Cost Whole-Body Teleoperation"
_robot-learning.org/CoRL/2024/Conference — CoRL 2024_

### Official Review · Reviewer_JTSs · 2024-07-12
**The paper is well-written and easy to follow, especially with extensive hardware evaluations. An additional contribution of this paper to the robotics community is its intent to open-sourcing the system. Discussions on imitation learning involving heterogeneous robotic systems would be beneficial.**

**Originality:** 4
**Technical Quality:** 5
**Clarity Of Presentation:** 4
**Potential Impact:** 3
**Recommendation:** 3
**Confidence:** 4

**Review:**

**Strengths:**

This paper is well-written and easy to follow with ample references. The paper includes detailed descriptions of the proposed hardware platform and experimental results. Supplemental videos demonstrate the effectiveness and robustness of the proposed learning framework in hardware experiments using the proposed platform. In addition, the paper aims to open source the system design. Open-sourcing the system could be potentially beneficial for the robotics community for advancement of future research.


**Weaknesses:**

In the current design, joint positions are directly used as a part of the mobile manipulation policy. This is possible because the leader and the follower arms share identical configurations. Thus, the collected datasets are applicable only in the specific platform. The paper uses the existing static ALOHA datasets for bimanual arm actions, but again this is possible since both static and mobile ALOHA systems have the common arm configurations. It would be informative if the authors discuss the case of imitation learning involving heterogeneous robotic systems.

**Minor comments:**

- page 3, Figure 2: Figure appears to be small, which could be enlarged for better readability.

- page 8, line 323: Reference (Zhao et al.) should be provided in the format with a number [1].

- The following paper (*1) conceptually employs a similar imitation learning approach to learning a mobile manipulation task with a teleoperated mobile manipulator platform, which could be mentioned in the paper.

(*1) Hiroshi Ito, Kenjiro Yamamoto, Hiroki Mori, and Tetsuya Ogata, "Efficient multitask learning with an embodied predictive model for door opening and entry with whole-body control", Science Robotics, Vol. 7, Issue 65, 2022.

- To make the paper be self-contained and more informative, brief explanations of the learning methods such as ACT, Diffusion Policy and VINN should be provided.

**Quality Of The Limitations Section:**

3

**Questions For Rebuttal:**

- How about the case of imitation learning involving heterogeneous robotic systems?
- Would it be possible to provide experimental videos at normal speed for teleoperation to better demonstrate the usability of the system?
- Can the learned policy generalize in the novel situations? For example, in the case of Wipe Wine task, can it work with the different table and glass?

**Robotics Focus:**

4

**Summary Of Paper:**

This paper presents Mobile ALOHA, a low-cost mobile manipulation platform for data collection and imitation learning. In this system, the original ALOHA system is augmented with a mobile base and a whole-body interface, enabling imitation of bimanual mobile manipulation tasks. Through co-training with static ALOHA datasets, it is demonstrated that Mobile ALOHA can learn various complex mobile manipulation tasks with small number of demonstrations in hardware experiments.

**Summary Of Recommendation:**

Overall, the paper is well-written and easy to follow, especially with extensive hardware evaluations. The intent to open source the system would be beneficial to the robotics community. However, at this moment, generalization of the learning framework using heterogeneous robotic systems is not very clear.

---

### Official Review · Reviewer_31qG · 2024-07-18
**The paper presents Mobile ALOHA, a cost-effective, whole-body teleoperation system that improves bimanual mobile manipulation tasks through co-training with static datasets, enhancing performance and efficiency in complex tasks with minimal demonstrations.**

**Originality:** 2
**Technical Quality:** 3
**Clarity Of Presentation:** 3
**Potential Impact:** 3
**Recommendation:** 3
**Confidence:** 4

**Review:**

**Strengths**
- **Cost-Effective Solution**: The system is low-cost, making advanced robotic manipulation accessible to more research labs.
- **Improved Task Performance**: The co-training method significantly boosts the success rates of complex tasks with minimal demonstrations.
- **Open-Source Accessibility**: The paper commits to open-sourcing all hardware and software implementations, promoting transparency and further research.

**Weaknesses**
- **Limited Novelty of Technical Methods**: This work is largely an incremental improvement on the existing ALOHA system, with a primary focus on hardware system evaluation. It merges with the previous dataset and employs similar methods, which limits the overall novelty. The primary innovation is the hardware setup, rather than new technical methods or algorithms, potentially reducing the paper's impact.

- **Potential Generalization Issue**: For some evaluations, the robot seems to consistently start from the same condition or using the very similar pots or opening the same cabinets, raising concerns about the system's ability to generalize to different initial conditions or environments. For example, while the authors likely tested the robot with various chairs, the results might be really good too, but this paper doesn't present these results. For the contribution to mobile and bimanual manipulation to be robust, the paper needs to show these robustness and versatility.

- **Handling Partial Dataset with Zero-Padding**: The method of merging two datasets by simply zero-padding the base action and third view camera raises questions about its scalability and rigor in handling diverse tasks, ie the actions can be values of zeros by nature. This approach may not be sufficient for robustly integrating partial data from different sources, potentially leading to suboptimal performance in tasks requiring more complex data handling and integration strategies.

**Quality Of The Limitations Section:**

2

**Questions For Rebuttal:**

[Issue 1] **Generalization and Robustness in Different Conditions**: Does quantitative evaluation (success metrics) include the trials with randomness shown in the example videos? Can the authors provide more detailed results or experiments showing the robot's performance starting from varied initial conditions or handling different types of chairs, pots with different weights or disturbances from the environments (move the objects or add weights into the pot during execution) to better demonstrate its generalization capabilities?

[Issue 2] **Zero-Padding**: How do the authors justify the robustness and scalability of their zero-padding method for merging datasets? Are there any additional experiments or analyses that support the effectiveness of this approach in handling diverse tasks and partial data?

**Robotics Focus:**

4

**Summary Of Paper:**

The paper introduces Mobile ALOHA, a low-cost, whole-body teleoperation system designed for bimanual mobile manipulation tasks. By augmenting the existing ALOHA system with a mobile base, the system allows for the collection of comprehensive bimanual manipulation data through teleoperation. The researchers utilize this data to train robotic policies using supervised behavior cloning. Their key finding is that co-training with static ALOHA datasets significantly improves the performance and success rates of mobile manipulation tasks. The paper demonstrates that with only 50 demonstrations per task, Mobile ALOHA can autonomously complete complex tasks like cooking, cleaning, and using elevators. This approach not only reduces the cost and complexity of hardware but also leverages existing datasets to enhance learning efficiency and performance in mobile robotic manipulation. The study underscores the potential of co-training with diverse datasets to develop more capable and efficient robotic systems.

**Summary Of Recommendation:**

The paper introduces Mobile ALOHA, a low-cost, whole-body teleoperation system for bimanual mobile manipulation, demonstrating significant improvements in task performance through co-training with static datasets. While the system's affordability and open-source nature are strengths, there are concerns about its generalization capabilities and robustness in varied scenarios. Overall, while the hardware innovation is notable, the limited novelty in technical methods suggests that further demonstrations of robustness and scalability are needed to fully validate the approach.

---

### Official Review · Reviewer_68bJ · 2024-07-18
**Low-Cost Whole-Body Mobile Manipulation Robot Leveraging Co-Training on Tabletop Data**

**Originality:** 3
**Technical Quality:** 4
**Clarity Of Presentation:** 4
**Potential Impact:** 4
**Recommendation:** 3
**Confidence:** 5

**Review:**

**Strengths**

1. **Low Cost and Accessibility**: The robot's low-cost design makes it financially accessible for a wide range of research labs. This enables research in mobile manipulation by allowing more institutions to participate. The authors have documented the materials and design process, ensuring that other researchers can easily replicate and build upon their work.

2. **Teleoperation System for Data Collection**: The authors introduce a teleoperation system enables efficient gathering of training data. The show the effectiveness of the collected data in their experiments.

3. **High Success Rate with Minimal Demonstrations**: Achieving up to a 90% success rate with only 50 demonstrations is a significant accomplishment. This efficiency indicates the robustness of their learning algorithms and the effectiveness of the training data.

4. **Using ALOHA Data for Co-Training**: By incorporating data from ALOHA (825 episodes), the authors significantly improve the robot's performance. Co-training on this dataset allows the robot to learn a broader range of skills and adapt to various manipulation tasks more effectively. This is a very interesting finding for the community, showing that transferring skills across different tasks using demonstration data is feasible.

5. **Significant Performance Improvement**: The co-training method achieves a 34% absolute improvement over prior methods. This substantial gain shows the effectiveness of their approach and its potential impact on the field.

6. **Effective in Fine-Grained Manipulation Tasks**: The robot works well with fine-grained manipulation tasks, showing controlled movements capability. This capability is crucial for applications that require detailed and careful handling of objects.

7. **Continuous Evaluation and Robustness**: The authors do continuous evaluation through tasks such as pushing chairs. This continuous testing ensures that the robot can perform reliably on the task and shows that the results are reliable and not cherry-picked.

**Quality Of The Limitations Section:**

2

**Questions For Rebuttal:**

1. **Feasibility in Real Houses**: The robot's large size may limit its practicality in navigating real houses, where space constraints are common. Addressing this issue is crucial for real-world applications in home environments. Do authors have any thoughts or further plans on how to deploy this setup in indoor environments beyond labs?

2. **Costly Data Collection for Long-Horizon Tasks**: Collecting 50 trajectories for long-horizon mobile manipulation tasks can be time-consuming and labor-intensive. Clarification is needed on the time required to gather data for the longest trajectories and how this impacts overall performance and efficiency.

3. **Generalization to New Environments**: The authors should provide more information on how well the robot generalizes to new, unseen environments. Understanding the limits of its adaptability is essential for evaluating its real-world applicability.

4. **Robustness to Environmental Changes**: The robot's performance under varying conditions, such as changes in background, clutter lighting, and dynamic obstacles (such as humans walking around) needs further exploration. These factors can significantly impact the robot's ability to perform tasks reliably.

5. **Battery Life and Operational Duration**: Information on how long the robot can operate untethered on a single charge would help determine its suitability for extended tasks.

6. **Scaling Data Collection**: Given the high cost associated with data collection, what do authors think about potential methods for scaling up this process?

7. **Navigation Duration and Environment Observability**: The paper should specify the duration of navigation tasks and whether the environment is partially or fully observable. This information is crucial for understanding the robot's navigation capabilities and its ability to handle complex environments.

8. **Collision Frequency and Handling**: Information on the frequency of collisions and how the robot handles them would provide insights into its safety and reliability.

**Robotics Focus:**

4

**Summary Of Paper:**

The paper introduces a new low-cost whole-body robot designed for mobile manipulation. It demonstrates that co-training with a larger dataset of tabletop manipulation significantly enhances the performance of mobile manipulation tasks.

**Summary Of Recommendation:**

I recommend accepting the paper for publication with minor revisions. The paper introduces a low-cost whole-body mobile manipulation robot and demonstrates significant performance improvements through co-training on a larger dataset. Its strengths include hardware accessibility, high success rates with minimal demonstrations, and substantial performance gains. However, the paper would benefit from addressing the robot's feasibility in real-world home environments, its generalization to new settings, and its robustness to environmental changes.    **After Rebuttal**: I read the rebuttal and other reviews. I thank the authors for the clarifications and I still vote for the acceptance of the paper.

---

### Decision · Program_Chairs · 2024-09-04

**Decision:**

Accept

**Comment:**

The reviewers found many strengths in this submission, but also clearly articulated some questions for the rebuttal phase.
Thank you for your responses.